# Predictive Value of Baseline [18F]FDG PET/CT for Response to Systemic Therapy in Patients with Advanced Melanoma

**DOI:** 10.3390/jcm10214994

**Published:** 2021-10-27

**Authors:** Virginia Liberini, Marco Rubatto, Riccardo Mimmo, Roberto Passera, Francesco Ceci, Paolo Fava, Luca Tonella, Giulia Polverari, Adriana Lesca, Marilena Bellò, Vincenzo Arena, Simone Ribero, Pietro Quaglino, Désirée Deandreis

**Affiliations:** 1Department of Medical Science, Division of Nuclear Medicine, University of Turin, 10126 Torino, Italy; roberto.passera@unito.it (R.P.); giulia.polverari@unito.it (G.P.); adriana.lesca@unito.it (A.L.); marilena.bello@unito.it (M.B.); desiree.deandreis@unito.it (D.D.); 2Nuclear Medicine Department, S. Croce e Carle Hospital, 12100 Cuneo, Italy; 3Department of Medical Sciences, Section of Dermatology, University of Turin, C.so Dogliotti, 10126 Torino, Italy; marco.rubatto@unito.it (M.R.); paolo.fava@unito.it (P.F.); luca.tonella@unito.it (L.T.); simone.ribero@unito.it (S.R.); pietro.quaglino@unito.it (P.Q.); 4Department of Medical Science, University of Turin, 10126 Torino, Italy; riccardomimmo93@gmail.com; 5Division of Nuclear Medicine, IEO European Institute of Oncology IRCCS, 20141 Milan, Italy; francesco.ceci83@gmail.com; 6PET Center, Affidea IRMET, 10135 Torino, Italy; vincenzo.arena@affidea.it

**Keywords:** immune checkpoint inhibitors, PD-1, PD-L1, CTLA-4, immunotherapy, target therapy, BRAF, MET, melanoma, [18F]FDG PET/CT

## Abstract

Background/Aim: To evaluate the association between baseline [18F]FDG-PET/CT tumor burden parameters and disease progression rate after first-line target therapy or immunotherapy in advanced melanoma patients. Materials and Methods: Forty four melanoma patients, who underwent [18F]FDG-PET/CT before first-line target therapy (28/44) or immunotherapy (16/44), were retrospectively analyzed. Whole-body and per-district metabolic tumor volume (MTV) and total lesion glycolysis (TLG) were calculated. Therapy response was assessed according to RECIST 1.1 on CT scan at 3 (early) and 12 (late) months. PET parameters were compared using the Mann–Whitney test. Optimal cut-offs for predicting progression were defined using the ROC curve. PFS and OS were studied using Kaplan–Meier analysis. Results: Median (IQR) MTVwb and TLGwb were 13.1 mL and 72.4, respectively. Non-responder patients were 38/44, 26/28 and 12/16 at early evaluation, and 33/44, 21/28 and 12/16 at late evaluation in the whole-cohort, target, and immunotherapy subgroup, respectively. At late evaluation, MTVbone and TLGbone were higher in non-responders compared to responder patients (all *p* < 0.037) in the whole-cohort and target subgroup and MTVwb and TLGwb (all *p* < 0.022) in target subgroup. No significant differences were found for the immunotherapy subgroup. No metabolic parameters were able to predict PFS. Controversially, MTVlfn, TLGlfn, MTVsoft + lfn, TLGsoft + lfn, MTVwb and TLGwb were significantly associated (all *p* < 0.05) with OS in both the whole-cohort and target therapy subgroup. Conclusions: Higher values of whole-body and bone metabolic parameters were correlated with poorer outcome, while higher values of whole-body, lymph node and soft tissue metabolic parameters were correlated with OS.

## 1. Introduction

Cutaneous malignant melanoma (CMM) and mucosal melanoma (MM), both malignancies of melanocyte cells, have been strongly increasing in the last 40 years, with about 287,700 new cases each year (3.1 cases per 100,000 inhabitants-year) globally in the world [1,2,3,4]. These malignancies are characterized by high aggressiveness and poor prognosis. In particular, CMM is one of the most aggressive type of skin cancer and is still associated with poor outcome, causing 90% of skin cancer mortality [5].

Recently, the introduction of two major systemic therapies has revolutionized the treatment of advanced melanoma, reducing the mortality in treated metastatic melanoma patients: molecular targeted therapy and immunotherapy [6]. Target therapies are mainly based on the use of small molecule inhibitors for v-raf murine sarcoma viral oncogene homolog B1 (BRAF)- and/or mitogen-activated protein kinase (MEK)-mutated melanomas, which specifically inhibit the most common oncogenic driver mutation responsible for melanoma cell proliferation and survival [7,8,9]. On the other hand, immune checkpoint inhibitors (ICIs) are based on the use of monoclonal antibodies targeting immunomodulatory receptors such as cytotoxic T-lymphocyte-associated protein 4 (CTLA-4) or programmed cell death protein 1 (PD-1), administered alone or in combination. Anti-CTLA-4 therapies and anti-PD1 therapies are able to draw cytotoxic T cells onto tumor cells by blocking these inhibitory checkpoints of the immune system [2,6,10,11].

Despite the high impact of systemic therapies on melanoma patient outcomes, a significant percentage of patients still do not achieve a response or relapse after treatment. The reasons for response heterogeneity and tumor relapse are still not clear and optimal biomarkers to predict response assessment have not yet been identified [12,13,14,15]. A personalized approach to select the right therapy, to predict the response to treatment and to avoid unnecessary toxicities appears necessary. Indeed, beside the clinical extension of disease, several predictive biomarkers have already been explored, such as histopathological, circulating and clinical biomarkers, as well as immunological and molecular markers [6,16,17,18,19,20,21].

Since 2019, 18F-fluorodeoxyglucose ([18F]FDG) positron emission tomography/computed tomography (PET/CT) has gained a leading role in malignant (stage III and IV) melanoma staging, treatment response assessment and post treatment surveillance of these two major systemic therapies in melanoma [22], while its role as a predictor of response to therapy is still under investigation, with conflicting results in recent studies, requiring further investigation [9,23,24,25,26].

Semiquantitative parameters, such as the maximum standardized uptake value (SUVmax), have been widely explored in the oncology field. However, in recent years, interest has shifted more toward parameters that allow assessment of tumor metabolic burden (MTB) from [18F]FDG PET/CT images. MTB calculation is based on two PET parameters: metabolic tumor volume (MTV), which indicates the volume of metabolically active tumor, and total lesion glycolysis (TLG), which is the product of SUVmean and MTV (providing information about average total tumor glycolysis). Moreover, these new parameters provide both global and district assessment of MTB, allowing the evaluation of the prognostic weight of the involvement of certain districts, such as bone or liver [27].

Indeed, we aimed to investigate whether semiquantitative parameters and metabolic tumor parameters on baseline [18F]FDG PET/CT scans (after primary tumor surgery) are able to: 1—predict response to target- or immuno-therapy at early (three months) and late evaluation (twelve months) after initiation of systemic therapy in a cohort of metastatic melanoma patients; 2—predict progression-free survival (PFS) and overall survival (OS).

## 2. Materials and Methods

### 2.1. Patient Selection

This was a retrospective, observational, non-interventional, multicenter study, conducted at the AOU Città della salute e della Scienza di Torino.

We retrospectively analyzed a cohort of 236 metastatic melanoma patients, treated with either immunotherapy (checkpoint-inhibitors, such as anti-PD-1/anti-CTLA-4) or target therapies (BRAF and/or MEK inhibitors) between January 2018 and January 2020.

Only patients with a documented willingness for their medical data to be used for research were then included in this retrospective, observational study. The study was conducted in compliance with ICH-GCP rules and the Declaration of Helsinki and approved by the Institutional Ethics Committee of University of Turin as part of the project TESEO (“Traguardi di Eccellenza nelle Scienze mediche Esplorando le Omiche”-protocol code D15D18000410001).

Eligible patients matched all the following inclusion criteria: (a) histologically proven melanoma; (b) immunotherapy or target therapy administered as the first-line treatment; (c) pre-treatment [18F]FDG PET/CT (time-point 0, TP0) performed within four months prior to systemic therapies; (d) availability of baseline CT, first post-treatment CT (time-point 1, TP1) within 3 months after the start of systemic therapy and a second post-treatment CT (time-point 2, TP2) within 12 months after the start of systemic therapy.

Exclusion criteria were: (a) they were under 18 years of age; (b) lack of follow-up/baseline imaging and clinical data; (c) patients with others concomitant oncological pathology; (d) patients treated with previous cycles of systemic therapies prior to undergoing study therapy; (e) patients enrolled for systemic therapy as neoadjuvant treatment.

### 2.2. Clinical Evaluation and Melanoma Classification

The following characteristics of the patients selected for the study were retrieved from the clinical database of the Dermatology Department of AOU Città della Salute e della Scienza: age, sex, genetic mutations, TNM stage at the time of PET scan, tumor staging according to AJCC VIIIth edition [28], previous, ongoing and during follow-up therapies.

### 2.3. PET/CT Acquisition

All included patients underwent a [18F]FDG PET/CT scan in a dedicated tomograph: -Philips Gemini Dual-slice EXP (Philips Medical Systems, Cleveland, OH, USA) at AOU Città della Salute e della Scienza;-Discovery 610 and Discovery IQ (GE Healthcare, Chicago, IL, USA) at Affidea-IRMET.

Patients were instructed to fast for at least 6 h before the scan, and blood glucose levels were measured before the injection of [18F]FDG. Patients were excluded if their blood glucose levels at the time of the scans exceeded 150 mg/dL (median (IQR) = 5.1 (4.7|6.9) mmol/L). The intravenous injected tracer activity was of 2.5–3 MBq/kg of 18F]FDG (median (IQR) = 230.0 (210.0|269.0) MBq), according to EAMN procedure guidelines [29].

After an uptake time of 60 min (median (IQR) = 73.0 (57.0|112.0) min) and following native low-dose CT acquisition both for attenuation correction and anatomical correlation (from the vertex of the skull to the feet), PET data were acquired, covering the identical anatomical region of the CT. The PET scans were reconstructed with ordered subset expectation maximization (OSEM) algorithms. The tomographs results were validated for a proper quantification and quality of the images recorded.

### 2.4. Quantitative Imaging Analysis

All PET/CT images were qualitatively analyzed with a dedicated workstation (Advantage; GE Healthcare) and were interpreted by one nuclear medicine physician (V.L.), aware of clinical data. Only lesions found to be metastatic on a combined assessment based on PET data, clinical analysis and follow-up were segmented and included in the semi-quantitative analysis.

For each metastatic lesion, PET semi-quantitative parameters were evaluated, including maximum standardized uptake value (SUVmax), metabolic tumor volume (MTV) and total lesion glycolysis (TLG). All FDG-avid lesions were semi-automatically segmented by one nuclear medicine physician using LIFEx v. 6.0 (IMIV/CEA, Orsay, France) [30].

LIFEx software provide the possibility to perform a semi-automatic whole-body MTV and TLG, performing the following steps:-process initialization by selecting regions with a standard uptake value (SUV) above a predefined threshold (SUV > 2.5) and applying a threshold set at 41% of maximum standard uptake value (SUVmax);-automatic calculation of the whole-body metabolic tumor volume (MTVwb) and total lesion glycolysis (TLGwb);-visual analysis of the resulting automated volume segmentation to remove background physiologic uptake.

For each lesion, the MTV and TLG were extracted. TLG was calculated by multiplying the MTV of each lesion with its corresponding SUVmean value. Each FDG-avid lesion with clear delineation of the tumor was used for MTB parameters calculation. Each volume of interest (VOI) has been classified according to its site including soft tissue (st), lymph node (ln), lung, liver, and bone.

At the end of this process, the whole-body MTV and TLG (MTVwb and TLGwb) was calculated, defined as the sum of all MTV and TLG of each lesion, respectively.

We also calculated the corresponding MTV (MTVst, MTVln, MTVlung, MTVliver and MTVbone) and TLG (TLGst, TLGln, TLGlung, TLGliver and TLGbone) according to each tumor site.

For all patients, fixed VOIs were drawn over the right lobe of the liver and aortic arch to evaluate blood pool and parenchymal organ background, respectively, measuring as mean standard uptake value (SUVmean).

### 2.5. Assessment of Therapy Response—Endpoints

Therapy response (TR) was routinely assessed on the level of individual lesions, using response evaluation criteria in solid tumors (RECIST) 1.1 criteria [31] and comparing lesion diameter at three different time-points based on contrast-enhanced (ce) CT images: baseline (TP0), first follow-up at 3 months (TP1), second follow-up at 12 months (TP2). When ce-CT evaluation was not feasible for fast disease progression or the worsening of patient clinical condition, the response to therapy was performed by clinical and laboratory evaluation only. Based on RECIST 1.1, the therapeutic response was assessed on ce-CT data only.

Regarding the immunotherapy sub-cohort, pseudo progression (PP) was defined as a diameter increase by ≥20% at TP1, followed by a decrease to <20% at TP2 compared to TP0. True progressive disease (TPD) was defined as an increase by ≥20% on both TP1 and TP2 compared to TP0.

Based on CT response assessment, patients were classified as “responder” in case of clinical benefit (including complete response, partial response, and stable disease) or “non-responder” in case of disease progression at first follow-up at 3 months (TP1) and second follow-up at 12 months (TP2), respectively.

Overall survival (OS) was defined as the time from treatment initiation to the date of death. Progression-free survival (PFS) was defined as the time from treatment initiation to the date of first progression/appearance of new lesions.

Study workflow is shown in Appendix A.

### 2.6. Statistical Analysis

Descriptive statistics for quantitative variables were expressed as median and interquartile range (IQR), while those for categorical variables as absolute/relative frequencies. The inferential tests were based on the Mann–Whitney test for continuous variables and the Fisher’s exact test for categorical ones.

For PET parameters (whole-body and districts MTV and TLG), we used a ROC analysis to determine the best cut-off allowing the prediction of the patient’s outcome using the Youden index [20]. The area under the curves (AUC), sensitivity, specificity and accuracy were reported.

The survival analysis was carried out using the Kaplan–Meier curves and the log-rank test to compare PFS and OS between the two groups (“responders” versus “non-responders”).

Statistical significance was considered for *p* < 0.05. Statistical analyses were performed using IBM SPSS version 26.0 (IBM, Armonk, NY, USA) [32].

## 3. Results

### 3.1. Patient and Primary Tumor Characteristics

Out of the 236 malignant melanoma patients who had undergone FDG PET/CT (retrospectively analyzed from the dermatology department database of our University Center), a total of 44 patients naive to systemic therapies were included in the final analysis (refer to CONSORT diagram, Figure 1).

Before PET/CT, all 44 patients (28M, 16F) had already undergone surgery (primary tumor and/or lymph node surgery). Of the 44 patients, 28/44 (63.6%) underwent target therapy and 16/44 (36.4%) underwent immunotherapy. Patient and tumor characteristics are listed in Table 1.

### 3.2. Semi-Quantitative PET Images Results—Metabolic Tumor Burden

The median (IQR) MTVwb was 13.1(4.3|30.9) mL in the entire cohort, 12.1(3.9|28.6) mL in the subgroup of patients treated with target therapy and 14.9(5.4|46.2) mL in the subgroup of patients treated with immunotherapy. The median (IQR) TLGwb was 72.4 (11.6|17.5) in the entire cohort, 65.2(10.2|257.8) in the subgroup of patients treated with target therapy and 94.6 (22.7|263.6) in the subgroup of patients treated with immunotherapy. Details of the semi-quantitative parameters extrapolated for each district from the PET images are summarized in Appendix A.

As shown in Figure 2, PET/CT images showed soft tissue, lymph node, lung, liver, and bone involvement in, respectively, 25.0%, 70.5%, 38.6%, 6.8% and 13.6% of cases in the whole cohort; in 28.6%, 78.6%, 17.9%, 7.1% and 14.3% of cases, respectively, in the target therapy subgroup; in 18.9%, 56.3%, 75%, 6.3% and 12.5% of cases in, respectively, in the immunotherapy subgroup.

When stratifying patients according to stage at PET scan time (11/44 stage III and 33/44 stage IV), in the entire cohort, the median (IQR) MTVwb and TLGwb was 4.3(2.6|30.2) mL and 12.3(5.3|408.2), respectively, for the stage III, and 16.0(7.3|33.6) mL and 79.1(28.8|225.8), respectively, for the stage IV. However, these differences were not statistically significant (*p* = 0.178 for MTVwb and *p* = 0.406 for TLGwb).

### 3.3. Early and Late Response Assessment

According to RECIST 1.1 criteria, the favorable outcome (complete response + partial response + stable disease = ‘patients with clinical benefit’) in the entire cohort, in the target therapy subgroup and in the immunotherapy subgroup was, respectively, identified in 6/44 (13.6%), 2/28 (7.1%) and 4/16 (25%) of cases at 3 months and in 11/44 (25%), 7/28 (25%) and 4/16 (25%) of cases at 12 months. For the immunotherapy subgroup, no pseudoprogression events occurred at 3 months, as all progressions at 3 months were confirmed at 12 months.

On the Mann–Whitney test, no semi-quantitative PET parameter was associated with response to therapy at 3 months in either in the whole-cohort or the two subgroups.

Conversely, at the 12-month evaluation, higher values of skeletal metabolic tumor burden in the entire cohort and in the target therapy subgroup (MTVbone, TLGbone and SUVmax-bone; all *p* < 0.037) and higher values of total metabolic tumor burden (MTVwb and TLGwb; all *p* < 0.022) in the target therapy subgroup were associated with radiological disease progression (non-responder patients). Data are summarized in Appendix A.

For the above-mentioned parameters, optimal cut-offs to predict responder vs. non-responder patients at 12 months were defined using the receiver operating characteristic (ROC) curve, the results of which are shown in Table 2 and Appendix A.

### 3.4. Patients’ Outcome Results

Median (IQR) follow up for the whole cohort and for the target therapy and the immunotherapy sub-groups was 21.0(13.2|35.7), 22.5(15.0|37.7) and 16.5(7.5|34.7), respectively. In the entire cohort, there were 11/44 (25%) deaths and 22/44 (50%) progression events; OS was 24.2 (range: 2.0–59.0; IQR: 13.2–35.7), while PFS was 21.0 (range: 2.0–53.0; IQR: 10.2–32.7). In the target therapy sub-cohort, there were 6/28 (21.4%) deaths and 13/28 (46.4%) progression events; OS was 26.8 (range: 9.0–59.0; IQR: 15.0–37.7), while PFS was 22.6 (range: 3.0–53.0; IQR: 14.0–31.7). In the immunotherapy sub-cohort, there were 5/16 (31.3%) deaths and 9/16 (56.3%) progression events; OS was 19.6 (range: 2.0–42.0; IQR: 7.5–34.7), while PFS was 18.0 (range: 2.0–42.0; IQR: 3.2–33.7).

On the Mann–Whitney test, no semi-quantitative PET parameter was associated with progression (PFS) either in the whole cohort or in the two subgroups.

Conversely, the following semi-quantitative PET parameter was able to predict favorable versus poor outcome (OS): lymph nodes (MTVlfn, *p* = 0.011; TLGlfn, *p* = 0.005; SUVmax-lfn, *p* = 0.036), soft tissue + lymph nodes (MTVsoft + lfn, *p* = 0.036; TLGsoft + lfn, *p* = 0.018; SUVmax-soft + lfn; all *p* = 0.047) and whole-body (MTVwb, *p* = 0.044; TLGwb, *p* = 0.009; SUVmax-wb, *p* = 0.029) metabolic tumor burden in the entire cohort; lymph nodes (MTVlfn, *p* ≤ 0.020; TLGlfn, *p* = 0.005), soft tissue + lymph nodes (MTVsoft + lfn, *p* = 0.010; TLGsoft + lfn, *p* = 0.004) and whole-body (MTVwb, *p* = 0.045; TLGwb, *p* = 0.008) metabolic tumor burden in the target therapy subgroup. For the above-mentioned parameters, optimal cut-offs to predict OS were defined using the receiver operating characteristic (ROC) curve, the results of which are shown in Table 3 and Appendix A.

In the entire cohort and target therapy subgroup, Kaplan–Meier curves showed a marginal trend in predicting disease progression among patients with MTVlfn, TLGlfn, MTVsoft + lfn, TLG soft + lfn, MTVwb and TLGwb lower or higher than the median values (Figure 3 for the entire cohort and Figure 4 for the target therapy cohort).

## 4. Discussion

Over the past decade, [18F]FDG PET/CT has gained a central role in stage III and IV melanoma staging and has become widely used in this clinical scenario, even if limited and heterogeneous data are currently available regarding the role of PET/CT semi-quantitative parameters as predictors of patient outcome [9,22].

Whole-body MTV and TLG are the PET-derived parameters that have been previously applied to assess the tumor metabolic activity in malignant melanoma patients [25,33,34]. To the best of our knowledge, this is the first study that attempts to identify PET parameters to assess their predictive value of disease progression and survival not only based on whole-body involvement (whole-body tumor metabolic burden), but also as district-based disease involvement.

The most significant results of our study were as follows: we observed (1) PET semi-quantitative parameters were not significantly correlated with radiological progression at three months; (2) there was a higher MTVbone and TLGbone value (all *p* < 0.037) in both the entire cohort and in the target therapy subgroup as well as higher MTVwb and TLGwb value (all *p* < 0.022) in the target therapy subgroup correlated with radiological progression at twelve months; (3) higher MTVlfn, TLGlfn, MTVsoft + lfn, TLGsoft + lfn, MTVwb and TLGwb value (all *p* < 0.05) in both the entire cohort and in the target therapy subgroup were correlated with OS; (4) no correlation was found between PET semi-quantitative parameters and both radiological progression and OS in patients receiving immunotherapy.

From these results, we can deduce that [18F]FDG baseline PET/CT may have a role in predicting disease progression in patients with high tumor burden, especially in patients receiving target therapy.

Compatible with the small sample size, our study suggests that the bone involvement could be a predictor of worse response to new generation systemic therapies. The skeleton is the fourth site of metastasis in malignant melanoma (occurring in about 11–18% of patients); however, the impact of bone disease in melanoma has been scarcely investigated [35]. However, our results appear to be in line with data collected from the SEER (Surveillance, Epidemiology and End Results) database, which showed how bone metastases are frequently associated with poor prognosis [36]. Recently, in their retrospective survey on bone metastasis in melanoma, Mannavola et al. [35] confirmed the unfavorable impact of bone metastases on patient survival. The authors found a direct correlation between prognosis and skeletal tumor burden (</> of 5 bone lesions); moreover, their analysis revealed that patients receiving ICIs and/or targeted agents showed a better prognosis (9.0–16.5 months) than those undergoing chemotherapy (4.0 months).

With regard to the results obtained in the immunotherapy subgroup patients, the small sample size certainly represents a limitation for the statistical analysis and future studies with the inclusion of a larger sample are needed to evaluate the possible association of PET district-based semi-quantitative parameters with outcome. However, although several studies have already shown a correlation between PET parameters (MTVwb and TLGwb) and outcome [37], our results may be in line with what has recently been shown by several works. Tumor burden is not the only factor impacting the response to immunotherapy: indeed tumor microenvironments and the patient’s immune response seem to play a predominant role [11], also combined blood biomarkers (such as LDH + S100) and non-invasive 18F]FDG PET/CT-based radiomic models seem to be promising biomarkers for early differentiation of pseudo-progression [25]. Finally, the degree of T cell infiltration in the tumor environment has a pivotal role in understanding the tumor response to immunotherapy (since ICI are supposed to trigger the T cells activity) [16,17].

Hence, the incorporation of circulating biomarkers in predictive models of immunotherapy response has become crucial.

Regarding the analysis of overall survival, significant correlations were found in the stratification of the analyzed semi-quantitative parameters. Patients with higher metabolic tumor burden (MTVwb > 13.1 mL and TLGwb > 72.4 in the entire cohort; MTVwb > 12.1 mL and TLGwb > 65.2 in the target therapy subgroup) showed a worse prognosis.

These results agree with those found in the previous works of Seban et al. [33] and Ito et al. [38] performed, respectively, on 56 and 142 patients with malignant melanoma undergoing immunotherapy; although in both studies the median cut-off value identified was higher for MTV (25 mL and 26.8 mL, respectively, versus 13.1 mL), higher for TLG in the study of Seban et al. (TLG cut-off = 258) and comparable with the study of Ito et al. (TLG cut-off = 78.7 versus 72.4 at our study). These differences could be justified both by the limited sample of our study and by the different setting of patients evaluated (both target and immunotherapy in our study, only immunotherapy in the other two studies mentioned).

Finally, probably the most interesting finding in our work relates to the impact of lymph node and “soft tissue + lymph node” tumor burden on overall survival in both the entire cohort (MTV = 5.6 mL and 6.5 mL, respectively; TLG = 17.9 and 28.8, respectively) and in the target therapy subgroup (MTV = 6.4 mL and 8.6 mL, respectively; TLG = 38.0 and 42.5, respectively).

In 2015, Beasley et al. [39] reported a 5-year survival rate of 59% in patients without regional lymph node disease compared to 19% for those with lymph node disease. In addition, cutaneous and subcutaneous metastases of melanoma are known to be associated with the development of lymph node and/or systemic metastases [40,41] and with poor prognosis [42,43,44,45,46,47,48]. Knowing in which patients the lymph node and soft tissue tumor burden assessed at baseline [18F]FDG PET/CT scan may be associated with a worse prognosis could be extremely useful. This is especially the case for patients who are candidates for target therapy, as acquired resistance could eventually develop in most patients due to several secondary events including mutations that evolve in response to treatment [9]; it is important to recognize that development of new cutaneous lesions with high FDG uptake can reflect accelerated growth.

### Limitations

This work is not exempt from limitations. Due to the retrospective design of the study, the population selected was not homogeneous (e.g., different sample size between target and immunotherapy subgroup, range of time between PET scan and start of systemic therapy). However, this is a real-world scenario cohort, represented by patients who generally are referred to [18F]FDG PET/CT before systemic therapy in daily clinical practice.

This analysis was performed in a relatively small sample size. A larger cohort would be preferable. In our study, we observed an association of MTV and TLG with both responder vs. non-responder status and OS. Nevertheless, this association was not maintained in the Kaplan–Meier analysis and, despite a non-negligible trend, MTV and TLG were not statistically significantly associated with PFS. The limited number of events at the end of follow-up influenced the correlation between these metabolic parameters and responder status and OS. It is probable that our statistical model could become more robust in a larger cohort, which would also allow stratification of patients according to clinical stage and risk of progression (based on clinical, laboratory and radiological data).

## 5. Conclusions

Our study, although far from definitive, encouraged a consideration of the importance of the [18F]FDG PET/CT whole-body tumor metabolic burden, together with single-district involvement of metabolically active disease.

More specifically, skeletal involvement can negatively influence the outcome of systemic therapy and bone metabolic parameters could have a predictive role for disease progression, while lymph node and soft tissue metabolic parameters could have a predictive role for the OS. These results seem particularly useful in the evaluation of patients who are candidates for target therapy, where the interaction with the immune system seems to have less impact on disease evolution.

These data should in future be validated in prospective studies with a larger patient population.

## Figures and Tables

**Figure 1 jcm-10-04994-f001:**
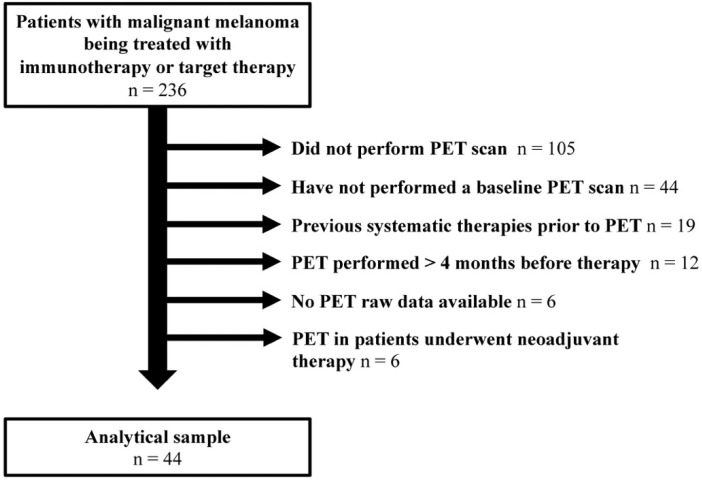
CONSORT DIAGRAM of patient inclusion/exclusion.

**Figure 2 jcm-10-04994-f002:**
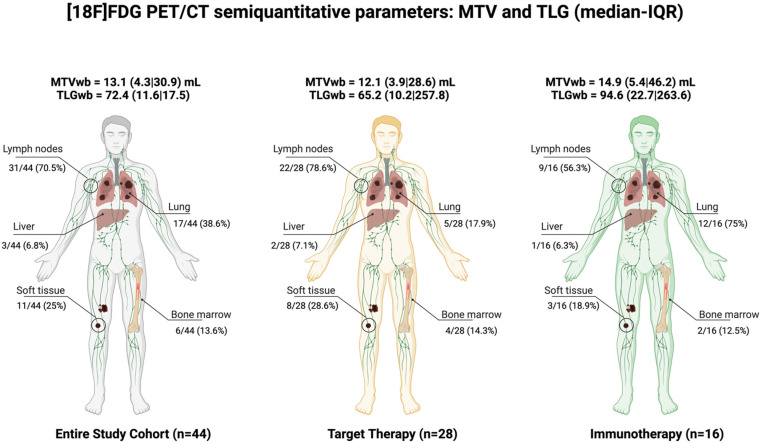
MTV and TLG whole-body and metabolic active lesions per district (interest expressed in percentage) for the entire cohort, the target therapy subgroup, and the immunotherapy subgroup, respectively.

**Figure 3 jcm-10-04994-f003:**
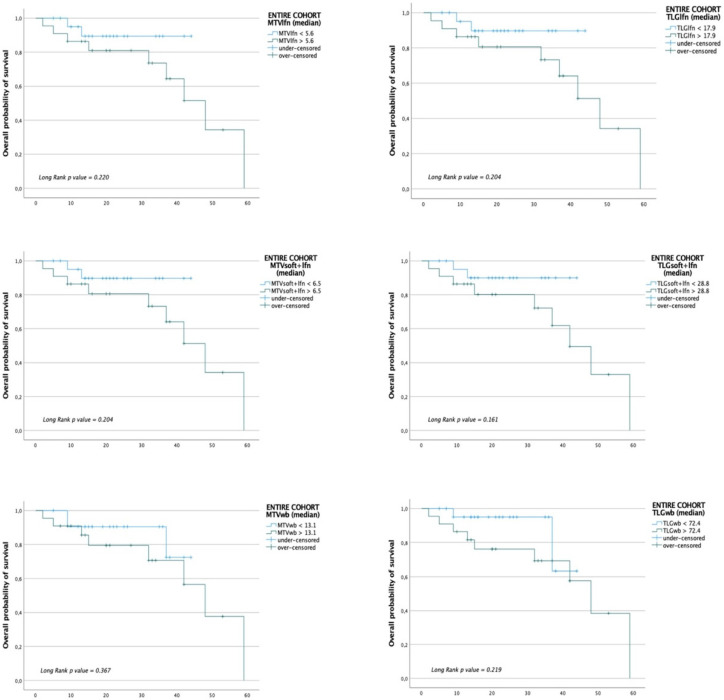
Kaplan–Meier plot analysis for whole-body and district MTV and TLG with overall survival (OS) in the entire cohort. Population was grouped by the median value.

**Figure 4 jcm-10-04994-f004:**
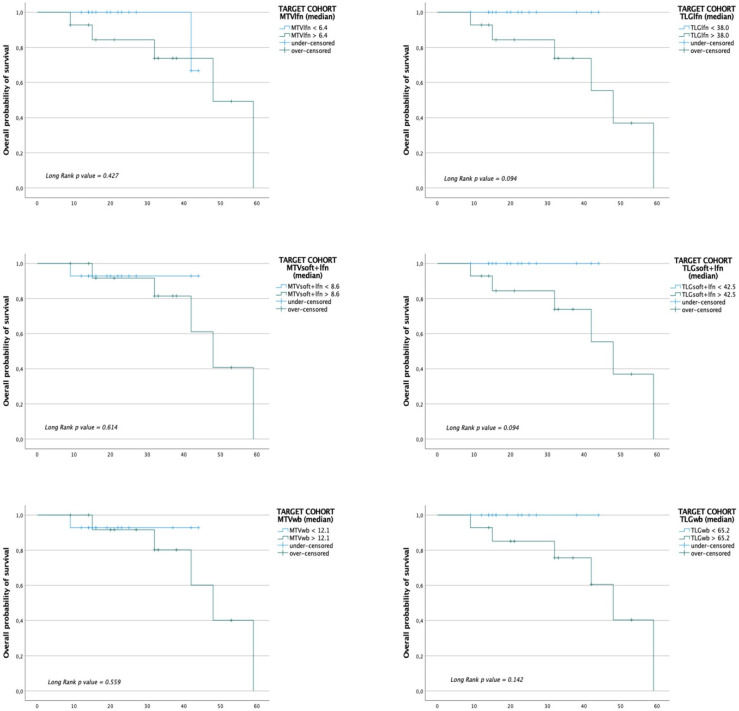
Kaplan–Meier plot analysis for whole-body and district MTV and TLG with overall survival (OS) in the target cohort. Population was grouped by the median value.

**Table 1 jcm-10-04994-t001:** Patient and primary tumor characteristics.

Patient Characteristics	
Gender, *n* (%)	
Male	28.0 (63.6)
Female	16.0 (36.4)
Age (years), median (IQR)	62.0 (49.7–75.0)
Primary melanoma characteristics	
Type, *n* (%)	
Superficial spreading melanoma (SSM)	12 (27.3)
Lentigo malignant melanoma (LMM)	2 (4.5)
Acral lentiginous melanoma (ALM)	0 (0.0)
Nodular melanoma (NM)	16 (36.4)
Unknown	8 (18.2)
Location, *n* (%)	
Head and neck	9 (20.4)
Torso	19 (43.3)
Limbs	10 (22.7)
Unknown	6 (13.6)
PET stage, *n* (%)	
III	11 (25.0)
IV	33 (75.0)
Breslow (mm), median (IQR)	4.5 (2.0–5.0)
Ulceration, *n* (%)	
Yes	8 (18.2)
No	11 (25.0)
Unknown	25 (56.8)
BRAF mutation, *n* (%)	
Yes	29 (65.9)
No	15 (34.1)

Note: BRAF = v-Raf murine sarcoma viral oncogene homolog B; MBq = megabecquerel; PET = positron emission tomography.

**Table 2 jcm-10-04994-t002:** Optimal cut-offs of semi-quantitative parameters associated with responder vs. non-responder patients at 12 months defined using the receiver operating characteristic (ROC) curve.

Parameters at 12 Months	Optimal Cut-Off	Sensitivity	Specificity	AUC	*p*-Value
Entire cohort					
Bone MTV (mL)	6.1	36%	97%	0.713	0.037
Bone TLG	18.8	45%	97%	0.716	0.034
Bone SUVmax	4.1	36%	97%	0.716	0.034
Target therapy cohort					
Bone MTV (mL)	13.1	42%	100%	0.786	0.027
Bone TLG	18.8	57%	100%	0.786	0.027
Bone SUVmax	5.7	42%	100%	0.786	0.027
Target therapy cohort					
Whole-body MTV (mL)	24.6	71%	85%	0.814	0.015
Whole-body TLG	208.4	71%	85%	0.793	0.023

**Table 3 jcm-10-04994-t003:** Optimal cut-offs of semi-quantitative parameters associated with OS defined using the receiver operating characteristic (ROC) curve.

Parameters for OS	Optimal Cut-Off	Sensitivity	Specificity	AUC	*p*-Value
Entire cohort					
Lymph nodes MTV (mL)	10.6	63%	76%	0.755	0.012
Lymph nodes TLG	55.1	63%	76%	0.777	0.006
Lymph nodes SUVmax	9.7	72%	64%	0.713	0.036
Entire cohort					
Soft tissue + LFN MTV (mL)	10.6	72%	67%	0.713	0.036
Soft tissue + LFN TLG	66.1	63%	70%	0.738	0.019
Soft tissue + LFN SUVmax	9.2	81%	61%	0.702	0.046
Entire cohort					
Whole-body MTV (mL)	14.8	72%	61%	0.705	0.043
Whole-body TLG	86.4	72%	64%	0.76	0.01
Whole-body SUVmax	11.6	72%	61%	0.72	0.03
Target therapy cohort					
Lymph nodes MTV (mL)	10.9	66%	73%	0.811	0.022
Lymph nodes TLG	137.4	66%	82%	0.864	0.007
Target therapy cohort					
Soft tissue + LFN MTV (mL)	14.6	66%	82%	0.841	0.012
Soft tissue + LFN TLG	132.1	66%	78%	0.871	0.006
Target therapy cohort					
Whole-body MTV (mL)	17.6	66%	69%	0.773	0.044
Whole-body TLG	158.1	66%	78%	0.848	0.01

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
