# Peer review of "Predictive Value of Baseline [18F]FDG PET/CT for Response to Systemic Therapy in Patients with Advanced Melanoma"

_jcm, 2021, doi:10.3390/jcm10214994_

Round 1
Reviewer 1 Report
Dear authors,
Thank you for giving us the opportunity to review your investigations. It it with interest that I went through your interesting manuscript.
Introduction:
69-72: According to the European guidelines 2019 for melanoma FDG-PET/CT has been established as tool in clinical practice for staging, treatment response assessment and post treatment surveillance. Response prediction has been investigated in recent studies with conflicting results requiring further investigation, as quoted further in your introduction. Therefore I would suggest to rephrase your sentence 71.
82-86: I assume the analysis was metastasis- and not primary tumor-based. So I would suggest to rephrase or describe that the parameters were "metastatic" (not based on the primary tumor).
Methods:
126-131: Did you use contrast medium? Native only? Or contrast for ocular melanoma only? I would suggest to be more specific here with regards on contrast medium.
Also Response assessment:
162: You stated RECIST 1.1. criteria were used. Were the lesions delineated on CT scans only or fused images? By the same physician?
I notice, you used the word "lesion" through your manuscript. Given the significant number of lymph nodes as "lesions" at baseline in your entire cohort and considering the specificity of FDG-PET/CT towards lymph nodes I think it would be interesting to report whether some of the lymph nodes at baseline came out false positive through clinical/imaging follow up or histopathology.
Results:
I would suggest to be more specific referring to OS, e.g. using either "favorable" or "poorer outcome".
Discussion:
I personally found the discussion a bit too long, with lack of strong focus and therefore sometimes confusing. I would suggest to report and discuss your key points on a lesion- and patient-level with focus on the patient-level, since the analysis/discussion on a patient-level is clinically more relevant.
I would also recommend to give your key point (4) 300-301 more credit, which I believe could be a key message of your manuscript for the group ICI , e.g. any consequence in clinical practice with patients under ICI if PET/CT parameters are not correlated with rad. progression and OS?
321: I would suggest not to refer to these results as negative. In fact, they are very interesting to discuss, as previously mentioned.
Conclusions:
In my opinion, the sentences are too long and therefore sometimes misleading.
388: I would suggest to specify on what level the prediction can be made (or not) and so summarize your results.
I would also recommend to go through the manuscript and correct spelling mistakes, e.g. physician vs. physicianS (136) or rich vs. reach (383). Furthermore, some sentences are too long and should be shortened for a better understanding.
I am looking forward to reading the revised version of your manuscript.
Best wishes and good luck!
Author Response
Dear Editor,
We would thank you for giving us the chance to revise and improve our manuscript. We also thank the reviewers for their useful comments.
We have provided a point-by-point rebuttal according to reviewer’s comments. All points raised by the reviewer have been evaluated, and the requested changes are included in the revised version of the manuscript, highlighted in red font in the text.
On behalf of all authors,
Virginia Liberini, MD
Reviewer #1:
Introduction:
69-72: According to the European guidelines 2019 for melanoma FDG-PET/CT has been established as tool in clinical practice for staging, treatment response assessment and post treatment surveillance. Response prediction has been investigated in recent studies with conflicting results requiring further investigation, as quoted further in your introduction. Therefore, I would suggest rephrasing your sentence.
A: We thank the reviewer for his/her comment, we have amended the sentence in accordance with the valuable suggestion (page 2, line 79-84).
82-86: I assume the analysis was metastasis- and not primary tumor-based. So, I would suggest rephrasing or describe that the parameters were "metastatic" (not based on the primary tumor).
A: We thank the reviewer for his/her comment, we have added a sentence “(after primary tumor surgery)” in the introduction section (page 2, line 95) and better clarify it in the methods section (page 4, line 154). Moreover, it was also stated in the result section (page 4 line 223-226): “Before PET/CT, all 44 patients (28M, 16F) had already undergone surgery (primary tumor and/or lymph node surgery)”.
Methods:
126-131: Did you use contrast medium? Native only? Or contrast for ocular melanoma only? I would suggest being more specific here with regards on contrast medium.
A: We thank the reviewer for his/her comment, we clarified this aspect in the methods section (page 3, line 145). We used only native low-dose CT for PET/CT acquisition. In our cohort, there was no ocular melanoma patient.
Also, Response assessment:
162: You stated RECIST 1.1. criteria were used. Were the lesions delineated on CT scans only or fused images? By the same physician?
A: We thank the reviewer for his/her comment, we clarified this aspect in the methods section (page 4, line 179-186). RECIST 1.1. were provided only on ce-CT images, data were extracted from the TESEO database, of which this study is a part of.
I notice, you used the word "lesion" through your manuscript.
Given the significant number of lymph nodes as "lesions" at baseline in your entire cohort and considering the specificity of FDG-PET/CT towards lymph nodes I think it would be interesting to report whether some of the lymph nodes at baseline came out false positive through clinical/imaging follow up or histopathology.
A: We thank the reviewer for his/her interesting comment. To avoid including non-metastatic findings in the analysis, only lesions found to be metastatic on a combined assessment based on PET data, clinical analysis and follow-up were segmented and included in the semi-quantitative analysis. As we have, thanks to your valuable comment, clarified and stated now in the materials and methods section (page 4, line 151-155).
Results:
I would suggest being more specific referring to OS, e.g. using either "favorable" or "poorer outcome".
A: We thank the reviewer for his/her comment, we have added the suggested terms in the results section.
Discussion:
I personally found the discussion a bit too long, with lack of strong focus and therefore sometimes confusing. I would suggest reporting and discuss your key points on a lesion- and patient-level with focus on the patient-level, since the analysis/discussion on a patient-level is clinically more relevant.
I would also recommend to give your key point more credit, which I believe could be a key message of your manuscript for the group ICI, e.g. any consequence in clinical practice with patients under ICI if PET/CT parameters are not correlated with rad. progression and OS?
321: I would suggest not to refer to these results as negative. In fact, they are very interesting to discuss, as previously mentioned.
A: We thank the reviewer for his/her comment, we have partly modified the discussion in line with what was suggested by both reviewers. However, we have maintained all our comments that we consider important for the evaluation and discussion of our data. n fact, we think that the added value of our research compared to what is present in the literature to date, is indeed the district analysis (not per-patient or per-lesion) that we carried out and that these data should be well discussed to give the readers a proper starting point for future work based on a per-patient, per-lesion but also per-district evaluation on a PET based level. As suggested, in the conclusions we have better explained how, in our opinion, these results could have a prognostic impact on the management of patients with advanced melanoma in clinical practice.
Conclusions:
In my opinion, the sentences are too long and therefore sometimes misleading sometimes misleading.
388: I would suggest specifying on what level the prediction can be made (or not) and so summarize your results.
A: We thank the reviewer for his/her comment, we have partly modified the discussion in line with what was suggested by both reviewers.
I would also recommend going through the manuscript and correct spelling mistakes, e.g. physician vs. physicians (136) or rich vs. reach (383). Furthermore, some sentences are too long and should be shortened for a better understanding.
I am looking forward to reading the revised version of your manuscript.
A: We thank the reviewer for his/her comment, the manuscript has been re-evaluated and spelling mistakes have been corrected and some sentences have been shortened.
Reviewer 2 Report
An interesting small original study evaluating baseline FDG-PET CT to predict response to systemic therapy in metastatic melanoma, confirming already known results from medical literature. Although patient's sample was quite small, I found it very interesting.
In the abstract you state : "44 melanoma patients who underwent [18F]FDG-PET/CT before first-line target therapy (28/50) or immunotherapy (16/50) were retrospectively analyzed." the enrolled patients seem 50 instead of 44...please check.
In the article you talk about advanced melanoma; in the introduction, you only refer to cutaneous melanoma; A small paragraph referring to mucosal melanoma could be added; such as: "Mucosal melanomas are malignant primary tumors originating from melanocytes located in the mucosal membranes . These malignancies are characterized by high aggressiveness and poor prognosis" and add a citation such as : doi: 10.3390/medicina57040359. Alternatively, you could change the title specifically referring to cutaneous melanoma, and adding that only primary cutaneous melanoma where considered in the study (although some of the melanoma are of unknown origin, so solution number one seems more adapt to this case).
Author Response
Dear Editor,
We would thank you for giving us the chance to revise and improve our manuscript. We also thank the reviewers for their useful comments.
We have provided a point-by-point rebuttal according to reviewer’s comments. All points raised by the reviewer have been evaluated, and the requested changes are included in the revised version of the manuscript, highlighted in red font in the text.
On behalf of all authors,
Virginia Liberini, MD
Reviewer #2:
An interesting small original study evaluating baseline FDG-PET/CT to predict response to systemic therapy in metastatic melanoma, confirming already known results from medical literature. Although patient's sample was quite small, I found it very interesting.
A: We thank the reviewer for his/her positive evaluation of our manuscript.
In the abstract you state: "44 melanoma patients who underwent[18F]FDG-PET/CT before first-line target therapy (28/50) or immunotherapy (16/50) were retrospectively analyzed." the enrolled patients seem 50 instead of 44...please check.
A: We thank the reviewer for his/her comment, we have corrected the erroneously reported data.
In the article you talk about advanced melanoma; in the introduction, you only refer to cutaneous melanoma; A small paragraph referring to mucosal melanoma could be added, such as: "Mucosal melanomas are malignant primary tumors originating from melanocytes located in the mucosal membranes. These malignancies are characterized by high aggressiveness and poor prognosis" and add a citation such as: doi:10.3390/medicina57040359. Alternatively, you could change the title specifically referring to cutaneous melanoma and adding that only primary cutaneous melanoma were considered in the study (although some of the melanoma are of unknown origin, so solution number one seems more adapt to this case).
A: We thank the reviewer for his/her comment, we add the sentence and the reference as suggested.